# Timing and Outcomes of Noninvasive Ventilation in 307 ARDS COVID-19 Patients: An Observational Study in an Italian Third Level COVID-19 Hospital

**DOI:** 10.3390/medicina58081104

**Published:** 2022-08-15

**Authors:** Nardi Tetaj, Pierluca Piselli, Sara Zito, Giada De Angelis, Maria Cristina Marini, Dorotea Rubino, Ilaria Gaviano, Maria Vittoria Antonica, Elisabetta Agostini, Candido Porcelli, Giulia Valeria Stazi, Gabriele Garotto, Donatella Busso, Silvana Scarcia, Assunta Navarra, Claudia Cimaglia, Simone Topino, Fabio Iacomi, Alessandra D’Abramo, Carmela Pinnetti, Gina Gualano, Alessandro Capone, Alberta Villanacci, Andrea Antinori, Fabrizio Palmieri, Gianpiero D’Offizi, Stefania Ianniello, Fabrizio Taglietti, Paolo Campioni, Francesco Vaia, Emanuele Nicastri, Enrico Girardi, Luisa Marchioni

**Affiliations:** 1UOC Resuscitation, Intensive and Sub-Intensive Care, National Institute for Infectious Diseases IRCCS Lazzaro Spallanzani, 00149 Rome, Italy; 2Department of Epidemiology, National Institute for Infectious Diseases IRCCS Lazzaro Spallanzani, 00149 Rome, Italy; 3Clinical and Research Department of Infectious Diseases, National Institute for Infectious Diseases IRCCS Lazzaro Spallanzani, 00149 Rome, Italy; 4Respiratory Infectious Diseases Unit, National Institute for Infectious Diseases IRCCS Lazzaro Spallanzani, 00149 Rome, Italy; 5Department of Radiology and Diagnostic Imaging, National Institute for Infectious Diseases IRCCS Lazzaro Spallanzani, 00149 Rome, Italy; 6Health and General Direction, National Institute for Infectious Diseases IRCCS Lazzaro Spallanzani, 00149 Rome, Italy; 7Scientific Direction, National Institute for Infectious Diseases IRCCS Lazzaro Spallanzani, 00149 Rome, Italy

**Keywords:** COVID-19, noninvasive ventilation, intensive care unit, acute respiratory distress syndrome, ARDS, NIV failure, orotracheal intubation

## Abstract

*Background and Objectives*: Background: Coronavirus disease 2019 (COVID-19) is a novel cause of Acute Respiratory Distress Syndrome (ARDS). Noninvasive ventilation (NIV) is widely used in patients with ARDS across several etiologies. Indeed, with the increase of ARDS cases due to the COVID-19 pandemic, its use has grown significantly in hospital wards. However, there is a lack of evidence to support the efficacy of NIV in patients with COVID-19 ARDS. *Materials and Methods*: We conducted an observational cohort study including adult ARDS COVID-19 patients admitted in a third level COVID-center in Rome, Italy. The study analyzed the rate of NIV failure defined by the occurrence of orotracheal intubation and/or death within 28 days from starting NIV, its effectiveness, and the associated relative risk of death. The factors associated with the outcomes were identified through logistic regression analysis. *Results*: During the study period, a total of 942 COVID-19 patients were admitted to our hospital, of which 307 (32.5%) presented with ARDS at hospitalization. During hospitalization 224 (23.8%) were treated with NIV. NIV failure occurred in 84 (37.5%) patients. At 28 days from starting NIV, moderate and severe ARDS had five-fold and twenty-fold independent increased risk of NIV failure (adjusted odds ratio, aOR = 5.01, 95% CI 2.08–12.09, and 19.95, 95% CI 5.31–74.94), respectively, compared to patients with mild ARDS. A total of 128 patients (13.5%) were admitted to the Intensive Care Unit (ICU). At 28-day from ICU admission, intubated COVID-19 patients treated with early NIV had 40% lower mortality (aOR 0.60, 95% CI 0.25–1.46, *p* = 0.010) compared with patients that underwent orotracheal intubation without prior NIV. *Conclusions*: These findings show that NIV failure was independently correlated with the severity category of COVID-19 ARDS. The start of NIV in COVID-19 patients with mild ARDS (P/F > 200 mmHg) appears to increase NIV effectiveness and reduce the risk of orotracheal intubation and/or death. Moreover, early NIV (P/F > 200 mmHg) treatment seems to reduce the risk of ICU mortality at 28 days from ICU admission.

## 1. Introduction

COVID-19 is an important novel cause of acute respiratory distress syndrome (ARDS) [1,2]. ARDS severity is assessed by the degree of hypoxemia, quantified by the ratio of arterial partial pressure of oxygen (PaO_2_) to the fraction of inspired oxygen (FiO_2_) as per the Berlin criteria, which is strongly predictive of worsening survival; a P/F ratio of 300 to 200 is mild, 200 to 100 is moderate, and less than 100 represents severe ARDS with Positive end-expiratory pressure (PEEP) ≥ 5 [3].

COVID-19 patients with progressive disease should be monitored closely for worsening respiratory status, dyspnea, hypoxemia, and decreased PaO_2_/FiO_2_ (ratio of arterial partial pressure, PaO_2_, to fractional inspired oxygen, FiO_2_), and typically require incremental respiratory support [4,5]. During the COVID-19 pandemic, noninvasive ventilation has been extensively used to avoid intubation in ARDS COVID-19 patients and overload of Intensive Care Unit (ICU) admissions, especially in conditions of limited resources [6,7]. Noninvasive ventilation (NIV), including biphasic positive airway pressure (BiPAP) and continuous positive airway pressure (CPAP), was recommended in the pre-COVID-19 era for acute hypoxemic respiratory failure due to cardiogenic respiratory failure and ARDS [8,9]. NIV has been commonly used for the treatment of acute respiratory failure in COVID-19 patients, and can be used in the ordinary ward at the bedside and allows multiple and rapid adjustments of ventilatory parameters such as positive end-expiratory pressure (PEEP), pressure support (PS), and fraction of inspired oxygen (FiO_2_) [6,10]. So far, however, no clinical practice guidelines have been developed for COVID-19 acute respiratory failure. Although the benefit of NIV treatment in reducing the need for invasive mechanical ventilation has been shown in studies published recently in JAMA [9,11], the timing of starting NIV treatment, its safety, and the risk of NIV failure in these patients is still debated and unclear in the medical literature [6,12,13].

Hence the purpose of the current study is to analyze:-The factors associated with NIV failure defined by the occurrence of orotracheal intubation and/or death as a combined outcome within 28 days from NIV start.-The factors associated with the risk of death within 28 days from ICU admission in COVID-19 patients, treated with or without NIV prior to Oro-Tracheal Intubation (OTI).

## 2. Materials and Methods

### 2.1. Study Design and Participants

The study was conducted at the National Institute of Infectious Diseases Lazzaro Spallanzani, Rome, Italy, which is a third-level COVID-19 center with over 200 hospital beds for infectious diseases and up to 55 beds in ICU. This observational cohort study included adult ARDS COVID-19 patients hospitalized in our COVID center from 29 January 2020 to 30 September 2020.

To simplify the reading of this article we have assigned the common term “noninvasive ventilation” (NIV) to refer to both biphasic positive airway pressure (BiPAP) and continuous positive airway pressure (CPAP). Diagnosis of ARDS COVID-19 was performed based on thoracic CT scans and fulfilment of Berlin criteria [14].

Inclusion criteria were adult patients admitted into our hospital with COVID-19 pneumonia, and SARS-CoV-2 confirmed by nasal pharyngeal swab for reverse transcriptase polymerase chain reaction (rtPCR) assay.

Exclusion criteria were negative rtPCR for SARS-CoV-2 and no ARDS COVID-19 pneumonia diagnosed clinically and/or by radiological imaging.

The NIV group included all patients treated with noninvasive ventilation for more than 24 h.

All patients underwent chest CT scan or X-ray at hospital admission. The patients with diagnosed ARDS (fulfilling the Berlin criteria), dyspnea, hypoxemia and/or decreased PaO_2_/FiO_2_ were treated as a first step with conventional supplemental oxygen therapy, and then the support was gradually increased with noninvasive ventilation according to the patient’s condition.

All clinical decisions and management of the patients were performed by attending physicians, according to institutional protocols and regular practice. NIV was applied for patients with a worsening of respiratory failure represented by a PaO_2_/FiO_2_ ratio below 300 mmHg, a respiratory rate more than or equal to 25 breaths per minute, important lung involvement in radiological images, non-responsiveness to conventional supplemental oxygen therapy, and persistent low peripheral oxygen saturation, SpO_2_ < 92–94% with FiO_2_ > 60%. The scarce availability of beds during the waves of the pandemic did not always allow early NIV treatment of all COVID-19 patients in respiratory failure.

The NIV included continuous positive airway pressure (CPAP) via helmet and Boussignac mask, and bilevel positive airway pressure (BiPAP) via a noninvasive ventilator machine (Philips Respironics Trilogy). We did not use high-flow nasal oxygen therapy (HFNOT) during the study period, as the hospital was not equipped.

The first step of treating COVID-19 patients was 3–5 L/min oxygen administration through nasal googles or Venturi face mask, with FiO_2_ up to 60% targeting a SpO_2_ level ≥ 94%, and starting to monitor the vital signs. Patients with worsening respiratory status required an incremental respiratory support, starting with CPAP at PEEP 7.5 cm H_2_O and FiO_2_ such that SpO_2_ ≥ 94%. Alternatively, when CPAP devices were not available, or in cases of hypercapnic respiratory failure, risk of muscular exhaustion or further respiratory worsening, the support was switched to BiPAP starting with pressure support (PS) at 5 cmH_2_O with PEEP between 5 and 10 cmH_2_O, gradually increasing according to the patient’s response.

The main contraindications to NIV included inability to protect the airway (impaired swallowing and coughing), reduced consciousness, uncooperative patients, thoracic barotrauma, hemodynamic instability, multiorgan dysfunction, or facial or esophageal surgery.

The decision for ICU admission and/or OTI was made by physicians considering the sum of several parameters including hemodynamic instability, neurological deterioration, worsening respiratory failure despite noninvasive oxygen therapy, increased respiratory rate, and increased effort of the respiratory muscles and acidosis. Neither noninvasive or invasive ventilation, nor ICU admission was applied in patients who refused the treatment or were uncooperative. Patients were managed according to recommendations from published guidelines and good medical practice on protective ventilation, which specify optimal PEEP with a target peak inspiratory pressure (PIP) less than 30 cmH_2_O, and tidal volume 6–8 mL/kg of ideal body weight (IBW) in noninvasive ventilators, plateau pressure ≤ 30 cmH_2_O and driving pressure ≤ 15 cmH_2_O in invasive ventilators, to keep low the risk of barotrauma and self-inflicted lung injury (P-SILI) [15,16,17,18,19].

### 2.2. Data Collection

Data were collected for the ReCOVeRI project, a register of hospitalized COVID-19 patients since the beginning of the pandemic, for clinical research at the National Institute for Infectious Diseases L. Spallanzani IRCCS, approved by the internal Ethical Committee (decision number 164, 26 June 2020). The management of the registry has been adapted according the standards of EUnetHTA reported in the Registry Evaluation and Quality Standards Tool (EUnetHTA, 2019). All clinical decisions and management of the patients were performed by attending physicians, according to institutional protocols and regular practice. The collected data was first structured into an electronic dataset (a case report form-like, CRF) consisting of 5 main sections: laboratory data (laboratory); daily clinical data (daily); administered drugs (drugs); notification data (notification); and summary clinical data (clinical), following specific criteria and definitions. The data was then entered into a database created ad hoc according to the CRFs, based on a Microsoft SQL Server database, accessed by trained personal for data entry and management.

PaO_2_/FiO_2_ ratio (P/F) was categorized into four classes (>300, 200–299, 100–199, and <100) and was collected at hospital admission and at the start of NIV treatment. The comorbidities (arterial hypertension, cardiovascular diseases, diabetes, obesity (BMI > 30 kg/m^2^), kidney disease on stage 3–5 of CKD, COPD, neoplasm in the last 5 years, and chronic neurological disorders) were collected as dichotomous variables (yes/no). All patients gave informed consent for collecting personal data for research purposes.

### 2.3. Statistical Analysis

Quantitative variables were calculated as medians (interquartile range, IQR), while categorical variables were expressed as counts (N) and percentages (%). We compared the collected data of patients treated with NIV and without NIV. The statistical comparison was performed by means of the Mann–Whitney test for continuous variables and chi-square test (Fisher or chi-square test for trends where necessary) for categorical variables.

In order to identify the factors associated with NIV failure, the latter was defined as the necessity of orotracheal intubation or death within 28 days after the start of NIV treatment, considering all COVID-19 patients treated with NIV. Factors were identified through logistic regression analysis allowing the calculation of odds ratio (OR) and 95% confidence intervals, first in univariate analysis, and subsequently with multivariable logistic regression analysis selecting all potential cofounding factors through backward elimination, abolishing from the model all nonsignificant confounders (*p*-value > 0.10). The adjusted odds ratio (aOR) values and 95% CI were reported.

A similar approach was used for the second main objective, where we aimed to identify those factors associate with risk of death within 28-days of ICU admission.

All statistical analysis were performed using the statistical software SPSS version 27 (IBM Corp. IBM SPSS, Armonk, NY, USA).

## 3. Results

From 29 January 2020 to 30 September 2020, a total of 942 COVID-19 patients with pneumonia were admitted to our COVID hospital. Of these, 224 were treated with noninvasive ventilation (NIV) and 718 with conventional supplemental oxygen therapy (non-NIV) during hospitalization; of which groups 88 and 40 patients, respectively, were then admitted into the ICU. Therefore, a total of 128 patients were admitted to the ICU, as schematically shown in the flow-chart depicted in Figure 1.

### 3.1. Baseline Characteristics of NIV Patients

Demographics, comorbidities, and clinical course of the COVID-19 patients are shown in Table 1. Patients treated with NIV were more frequently male (*p* = 0.005), older (*p* = 0.048), had hypertension (*p* = 0.011), and were obese (*p* < 0.001). Chronic neurological disease was less frequent in treated patients (*p* = 0.006). Cardiovascular diseases, diabetes, chronic kidney disease, COPD, and neoplasm in the past five years showed no statistically significant differences between the two groups, as reported in Table 1.

Three hundred and seven COVID-19 patients fulfilled Berlin criteria for ARDS at hospital admission, of these, 141 were treated with NIV. Forty patients untreated with NIV were admitted into ICU and subsequently intubated. Three patients receiving NIV were intubated in the ward and then transferred to another hospital, Figure 1.

Before starting NIV, 218 of 224 patients had COVID-19 ARDS at hospital admission, of which 59 (26.3%) presented mild, 135 (60.3%) moderate, and 24 (10.7%) severe ARDS. The median time from hospital admission to NIV start was three days (IQR, 2–6).

A lower P/F ratio at hospitalization was significantly associated with NIV treatment, when compared between NIV and non-NIV groups, with a clear gradient of P/F ratio (*p* < 0.001).

The length of stay in hospital was significantly longer in patients treated with NIV (*p* < 0.001). Moreover, the patients that needed noninvasive ventilation had a higher rate of ICU admission (*p* < 0.001), need of orotracheal intubation (*p* < 0.001), and rate of death (*p* < 0.001) (see Table 1).

### 3.2. NIV Failure in COVID-19 Patients

Within 28 days from starting NIV, 86 patients were admitted into ICU, of which 63 were intubated, and 138 patients were not admitted in ICU, with 20 deaths occurring in this latter group, Figure 2. Table 2 shows the comparison between the 224 patients who underwent NIV and the 84 (37.5%) patients that failed NIV, defined by OTI requirement and/or death (combined outcome) within 28 days of follow-up from the start of noninvasive ventilation.

In the univariate analysis, factors significantly associated with NIV failure were: female, with a twice higher risk than male (OR 2.15, *p* = 0.013); older age, with a 62% risk increase (OR 1.62, *p* < 0.001) for each 10 years; hypertension, with a 2.6 times increased risk (OR 2.57, *p* = 0.001); COPD which conferred a 6.2-times higher risk (OR 6.23, *p* = 0.002), and history of neoplasm in the past five years, with a 3.2-fold higher risk (OR 3.17, *p* = 0.021). The presence of diabetes, neurological disease, obesity, or chronic renal diseases were not significantly associated with NIV failure.

The multiple regression analysis shows that female patients had a risk twice as high as male (aOR 2.12, *p* = 0.031). Also, age and COPD showed respectively and independently an increase in NIV failure of 25% for each 10 years (aOR = 1.25, *p* = 0.068) and about 3.5-fold (aOR = 3.44, *p* = 0.060), with a significance slightly above 0.05.

The overall rate of NIV failure was 37.5% (84 patients) in 224 NIV patients. NIV failure occurred in seven patients (10.8%) with P/F > 200 mmHg (mild ARDS), 58 (43%) with P/F 101–200 (moderate ARDS), and 19 (79.1%) when P/F ≤ 100 mmHg (severe ARDS). It is to be noted that of six patients with P/F > 300 mmHg at NIV start, none of them failed the treatment.

In the univariate analysis, a P/F ratio of 101–200 and ≤100 mmHg before NIV treatment had a six-fold (OR 6.24, *p* < 0.001) and 31.5-fold (OR 31.5, *p* < 0.001) increased risk of NIV failure, respectively, compared with patients presenting P/F ratio >200 mmHg. In the multiple regression analysis these risks became five-fold (aOR 5.01, *p* < 0.001) and 20-fold (aOR 19.95, *p* < 0.001) higher, respectively.

### 3.3. Risk of Death in COVID-19 Patietns Treated with NIV

Figure 3 and Table 3 show schematically the occurrence of death in ICU patients (27.3%) within 28 days of ICU admission. Sixty-one patients (47.7%) of those previously treated with NIV admitted into ICU were subsequently intubated, of which 17 (27.9%) died within 28 days of ICU admission. Further 27 patients treated with NIV and admitted into ICU did not require intubation, and only one patient, who repeatedly refused orotracheal intubation, died while still receiving NIV. The remaining 40 patients (31.3%) admitted into ICU and not previously treated with NIV were all intubated, and 17 (42.5%) died within 28 days of ICU admission. The mortality among intubated patients in ICU was 33.7% overall.

The univariate analysis identified factors significantly associated with the risk of death: chronic renal disease, with more than four-fold increased risk (OR 4.4, *p* = 0.011), hypertension, with a 2.4-fold greater risk (OR = 2.43, *p* = 0.031), and other cardiovascular diseases with a three-times greater risk (OR 3.13, *p* = 0.008). According to this analysis, female sex, age variation, COPD, diabetes, neurological disease, obesity, and history of neoplasm were not significantly associated with a different risk of death.

The multivariable regression analysis showed the cardiovascular disease was statistically significant, and independently associated with the outcome of death, with a 4.5-fold increased risk (aOR 4.56, *p* = 0.002).

Moreover, we observed that compared with the group who underwent OTI without prior NIV, those who were intubated after NIV treatment showed a tendency of decreased mortality risk. Interestingly, those admitted into ICU with previous NIV treatment who did not require orotracheal intubation showed a significant 95% reduction of mortality. The adjusted model also confirmed these findings, with a significant 94% reduction of mortality for those who did not require OTI.

## 4. Discussion

### 4.1. Baseline Characteristics of NIV Patients

Noninvasive ventilation in our hospital was a commonly used treatment for patients with severe COVID-19 pneumonia already in the hospital ward, while facing an overload of ICU beds. During the period of the study, 23.8% of all COVID-19 patients were treated with NIV.

COVID-19 patients who required NIV were more frequently male, older, and with more comorbidities such as arterial hypertension and obesity, compared with those never receiving NIV. Chronic neurological disease was less frequently found in patients treated with NIV; this could be explained by assuming that these patients due to their underlying disease (such as senile dementia, Alzheimer’s disease etc.) were less compliant and therefore more difficult to treat with noninvasive ventilation, which requires cooperation. Cardiovascular diseases, diabetes, chronic kidney disease, COPD, and history of neoplasm in the past five years were not shown to be significant predictors for NIV requirement.

At hospitalization, 307 (32.5%) COVID-19 patients fulfilled Berlin criteria for ARDS, with a statistically significant increased need for NIV treatment. In the non-NIV group, 53 patients with moderate-severe ARDS (P/F ratio ≤ 200 mmHg) had contraindications to NIV (uncooperative patients, thoracic barotrauma, hemodynamic instability, refused NIV, etc.) or they were already severe ill when they were admitted to hospital.

Indeed, 40 (5.6%) of the non-NIV group were admitted into ICU and subsequently intubated due to the worsening of their clinical conditions, of which 17 (42.5%) died within 28 days of ICU admission.

At the start of their NIV treatment, 218 patients (97.3% of NIV group) were diagnosed with ARDS (59 with mild, 135 with moderate, and 24 with severe ARDS). The length of stay in hospital was significantly longer in patients treated with NIV. Also, the patients that required noninvasive ventilation showed a higher rate of ICU admission, orotracheal intubation, and death compared with patients who did not require NIV.

### 4.2. NIV Failure in COVID-19 Patients

Among 224 NIV patients, NIV failure occurred in 84 (37.5%) patients, defined by the occurrence of intubation and/or death within 28 days of starting NIV. Patients that failed NIV were more frequently female, with twice the risk than male, older, and with more comorbidities including arterial hypertension, COPD, and neoplasm. Being female was the most significant predictor independently associated with a higher risk of NIV failure, older age and having COPD were slightly above the significance level, while other comorbidities such as diabetes, neurological disease, obesity, or chronic renal diseases were not found to be significantly associated with NIV failure.

The failure rate of noninvasive ventilation was independently correlated with the severity category of COVID-19 ARDS (PaO_2_/FiO_2_ ratio categories). Indeed, NIV failure occurred in 10.8% of COVID-19 patients with mild ARDS, 43% with moderate ARDS, and 79% of those with severe ARDS.

In 2016, pre-COVID-19, according to a large multicenter observational study LUNG SAFE, with data from 50 countries and 2,813 patients with ARDS, NIV failure occurred in 22.2% of mild, 42.3% of moderate and 47.1% of severe ARDS patients [20]. Therefore, from results of our study in comparison with non-COVID-19 ARDS patients in the cited study, we can observe that in mild ARDS patients (with P/F 200–299 mmHg) the risk of NIV failure was lower (10.8% vs. 22.2%), it was similar for moderate ARDS (for P/F 100–199 category, 43% vs. 42.3%), but became notably higher in cases of severe ARDS when the P/F ratio dropped below 100 mmHg (79.1% vs. 47.1%). Studies carried out during the COVID-19 outbreak show that noninvasive ventilation may have some benefit in postponing or avoiding worsened outcomes, but with limitations according to patient severity and PaO_2_/FiO_2_ value, although a smaller sample size was used [21,22].

Therefore, we observed that in COVID-19 patients with a P/F ratio > 200 mmHg the risk of NIV failure was low, indicating that an early start of NIV treatment at stages when COVID-19 patients have a P/F ratio > 200 mmHg increases the chance of NIV success, consequently avoiding orotracheal intubation. However, in the more advanced stages, especially when P/F falls below 100 mmHg, we should not hesitate to intubate these patients because they would be at very high risk of NIV failure.

### 4.3. Risk of Death in COVID-19 Patietns Treated with NIV

During the period of study, 13.6% of COVID-19 patients in the hospital ward were admitted into ICU. Of the population treated with NIV, 39.3% were admitted into ICU. Among those admitted to ICU, overall mortality within 28 days of ICU admission was slightly higher than one quarter (27.3%).

At 28-day follow-up from ICU admission, arterial hypertension, cardiovascular diseases, and chronic renal disease were significantly associated in the univariate regression analysis with an increased risk of death, but only cardiovascular disease with a 4.5-fold higher risk of death was found to be independently and significantly associated.

Mortality among those who were intubated in ICU was 33.7%. Patients treated with NIV who were subsequently admitted into ICU and underwent OTI had a mortality of 27.9%, was significantly lower than the 42.5% mortality of intubated patients who were not treated with prior NIV, at 28 days of ICU admission. One patient died while still in NIV, who had expressively given do-not-intubate (DNI) order.

These findings show that early noninvasive ventilation prior to OTI was independently associated with a significantly decreased risk of mortality in ICU.

Before drawing conclusions, several limitations should be considered. First, this is a retrospective observational study within a single third-level COVID-19 center, and as in any observational study controlling for confounders may be incomplete despite all efforts. Second, decisions about the timing and management of NIV treatment, OTI, and ICU admission were made by intensive care physicians based on institutional protocols and good medical practice. Therefore, we cannot rule out the possibility that physicians made decisions for NIV placement, OTI, or ICU admission based on pre-existing medical conditions (comorbidities, age, etc.), or the availability of medical devices and ICU beds, especially during multiple COVID-19 waves that overwhelmed the hospital. Additionally, patients who refused NIV, OTI, or ICU admission despite clinical severity, and who then died, may have generated an unmeasured bias that could underestimate the effectiveness of NIV. Third, we considered NIV treatment to include BiPAP and/or CPAP, without the possibility based on clinical charts of a reliable more accurate classification of the type of interface used for NIV, which may have been a potential determinant of the outcomes. Therefore, further research on larger prospective studies is required. However, this study also has certain strengths, such as its large sample size, and full representativeness of real-life clinical practice, including management of ICU COVID-19 patients with severe respiratory failure, which cannot be represented by a randomized controlled trial study. The findings of this study seem in line with some of the aforementioned research already presented in medical literature [6,12,13,20]. Observational retrospective studies are the first step to formulating hypothesis that could incentivize the design of larger prospective studies.

## 5. Conclusions

In conclusion, our study found that noninvasive ventilation was a useful mode of therapy in COVID-19 patients with ARDS. The study showed that NIV was effective in preventing a negative outcome (either need of intubation or death) in approximately two-thirds of patients with COVID-19 ARDS treated with this procedure.

NIV failure occurred in 37.5% of COVID-19 ARDS patients, in almost half of patients (43%) with moderate ARDS, and in about 80% of patients with severe ARDS, within 28 day of starting NIV. The presence of cardiovascular disease was an independent risk factor for increasing mortality.

Starting NIV at earlier stages of COVID-19, when P/F > 200 mmHg, was shown to be associated with a lower risk of NIV failure, avoiding orotracheal intubation and/or death.

Patients treated with NIV who were subsequently admitted into ICU and underwent OTI had significantly lower 28-day mortality after ICU admission, compared with intubated patients who were not previously treated with NIV. These findings suggest that NIV treatment is independently associated with a decreased risk of ICU mortality.

## Figures and Tables

**Figure 1 medicina-58-01104-f001:**
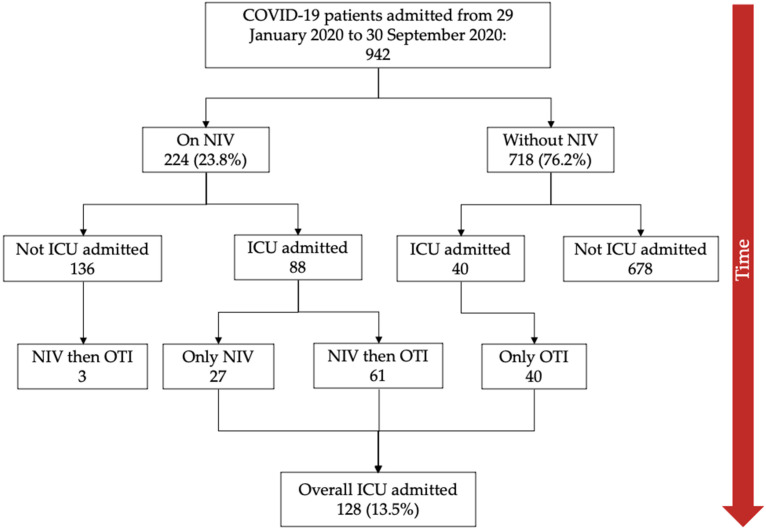
Flowchart of study selection. Abbreviations: COVID-19, coronavirus disease 2019; NIV, noninvasive ventilation; ICU, intensive care unit; OTI, orotracheal intubation.

**Figure 2 medicina-58-01104-f002:**
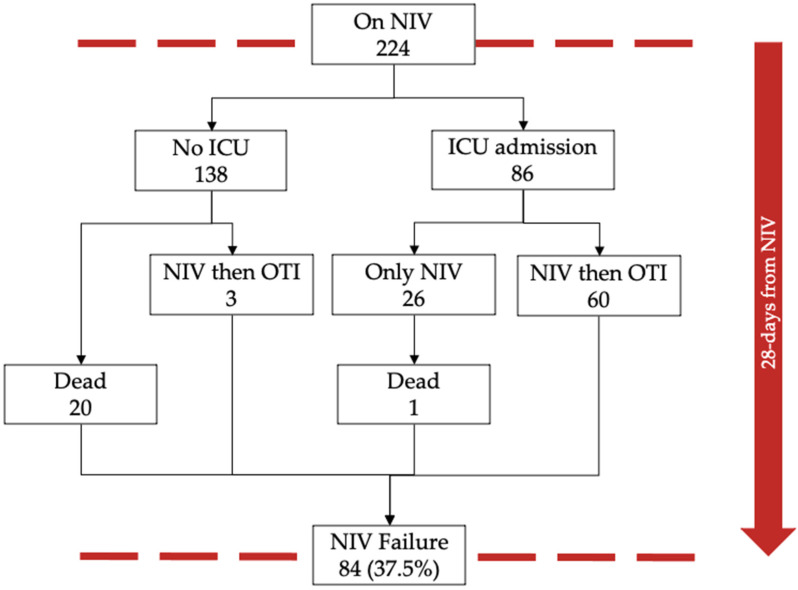
Flowchart of patients undergoing NIV. Abbreviations: NIV, noninvasive ventilation; ICU, intensive care unit; OTI, orotracheal intubation.

**Figure 3 medicina-58-01104-f003:**
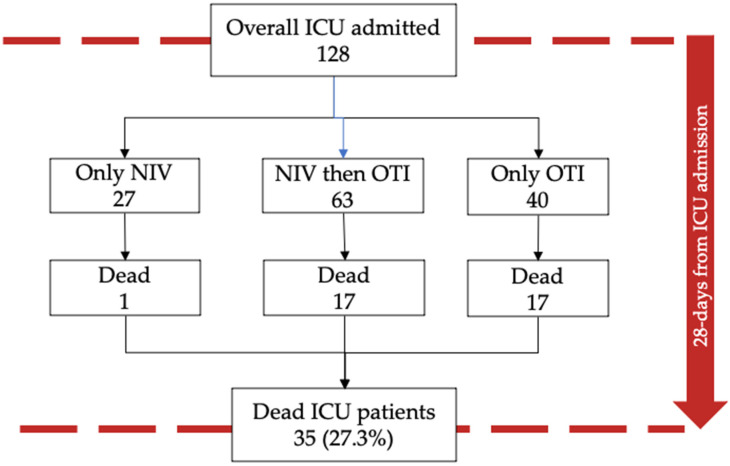
Flowchart of patients admitted into ICU. Abbreviations: NIV, noninvasive ventilation; ICU, intensive care unit; OTI, orotracheal intubation.

**Table 1 medicina-58-01104-t001:** Baseline demographics and clinical features of COVID-19 patients.

Characteristics	COVID-19 Patients	NIV Group	Non-NIV Group	*p* ^1^
	942	224	718	
Age, median (IQR)	60 (48–73)	63 (51–73)	59 (47–74)	**0.048**
Male, *n* (%)	612 (65)	163 (72.8)	449 (62.5)	**0.005**
Female, *n* (%)	330 (35)	61 (27.2)	269 (37.5)	
Comorbidities, no. (%)				
Arterial hypertension	357 (37.9)	101 (45.1)	256 (35.7)	**0.011**
Cardiovascular disease	236 (25.1)	62 (27.7)	174 (24.2)	0.299
Diabetes	168 (17.8)	49 (21.9)	119 (16.6)	*0.070*
Obesity **^a^**	143 (15.2)	58 (25.9)	85 (11.8)	**<0.001**
Chronic renal disease **^b^**	58 (6.2)	18 (8.0)	40 (5.6)	0.180
COPD	82 (8.7)	17 (7.6)	65 (9.1)	0.498
Neoplasm **^c^**	88 (9.3)	19 (8.5)	69 (9.6)	0.613
Chronic neurological disorders	132 (14.0)	19 (8.5)	113 (15.7)	**0.006**
ARDS patients at hospital admission, no. %	307 (32.5)	141 (63)	166 (23.1)	
PaO_2_/FiO_2_ at hospital admission				
>300	635 (67.4)	83 (37.1)	552 (76.9)	**<0.001**
201–300	187 (19.9)	74 (33.0)	113 (15.7)	
101–200	94 (10.0)	51 (22.8)	43 (6.0)	
≤100	26 (2.8)	16 (7.1)	10 (1.4)	
ARDS patients at starting NIV, no. %		218 (97.3)		
PaO_2_/FiO_2_ at starting NIV				
>300		6 (2.7)		
201–300		59 (26.3)		
101–200		135 (60.3)		
≤100		24 (10.7)		
Pre-NIV hospitalization, days, median (IQR)		3 (2–6)	N.A.	
Total length of stay, days, median (IQR)	15 (9–25)	26 (18–35)	13 (8–19)	**<0.001**
Overall follow-up, no. %				
Admitted in ICU	128 (13.6)	88 (39.3)	40 (5.6)	**<0.001**
Underwent OTI	104 (11.0)	64 (28.6)	40 (5.6)	**<0.001**
Death	100 (10.6)	43 (19.2)	57 (7.9)	**<0.001**
28-day follow-up from NIV, no. %				
Admitted in ICU		86 (38.4)		
Underwent OTI		63 (28.1)		
Dead		37 (16.5)		
OTI and/or death (combined variables)		84 (37.5)		

Abbreviations: IQR, interquartile range; COPD, chronic obstructive pulmonary disease; **^a^** obesity is defined as BMI > 30 kg/m^2^; **^b^** stage 3–5 of CKD, chronic kidney disease; **^c^** solid neoplasia or hematological malignancy in the past five years; ARDS, acute respiratory distress syndrome; ICU, intensive care unit; OTI, orotracheal intubation; **^1^** Chi square test was performed between the two groups. When *p*-value was <0.05 it was indicated in bold, while when <0.10 in italic.

**Table 2 medicina-58-01104-t002:** Unadjusted and adjusted predictors of NIV failure.

Characteristics	NIV Group	NIV Failure *no (%)	Univariable		Multivariable	
			OR (95% CI)	*p*	aOR (95% CI)	*p*
	224	84 (37.5)				
Male	163	53 (32.5)	1	**0.013**	1	**0.031**
Female	61	31 (50.8)	2.15 (1.18–3.91)		2.12 (1.07–4.20)	
Age (for 10 years of increase)			1.62 (1.31–2.01)	**<0.001**	1.25 (0.98–1.59)	*0.068*
Comorbidities						
Arterial hypertension	101	50 (49.5)	2.57 (1.47–4.47)	**0.001**		
Cardiovascular disease	62	29 (46.8)	1.71 (0.94–3.10)	*0.078*		
Diabetes	49	16 (32.6)	0.76 (0.39–1.49)	0.429		
Obesity **^a^**	58	25 (43.1)	1.37 (0.75–2.53)	0.307		
Chronic renal disease **^b^**	18	10 (55.5)	2.23 (0.84–5.9)	0.106		
COPD	17	13 (76.5)	6.23 (1.96–19.80)	**0.002**	3.44 (0.95–12.48)	*0.060*
Neoplasm **^c^**	19	12 (63.1)	3.17 (1.19–8.40)	**0.021**		
Neurological disorders	19	10 (52.6)	1.97 (0.77–5.06)	0.160		
PaO_2_/FiO_2_ at NIV						
>200	65	7 (10.8)	1	**<0.001**		**<0.001**
101–200	135	58 (43.0)	6.24 (2.65–14.68)		5.01 (2.08–12.09)	
≤100	24	19 (79.1)	31.49 (8.94–110.91)		19.95 (5.31–74.94)	

Abbreviations: * NIV failure is defined as the combined variable of orotracheal intubation or death at 28 days follow-up from starting NIV; COPD, chronic obstructive pulmonary disease; **^a^** obesity is defined as BMI > 30 kg/m^2^; **^b^** stage 3–5 of CKD, stages of chronic kidney disease; **^c^** solid neoplasia or hematological malignancy in the past five years; NIV, noninvasive ventilation; OR, odds ratio; aOR, adjusted odds ratio. When *p*-value was <0.05 it was indicated in bold, while when <0.10 in italic.

**Table 3 medicina-58-01104-t003:** Multifactorial analysis of the factors associated with an outcome of death.

COVID-19 Patients	ICU Patients	Dead ICU Patients *	Univariable		Multivariable	
No.	128	35 (27.3)	OR (95% CI)	*p*	aOR (95% CI)	*p*
Male, *n* (%)	87	24 (27.6)	1			
Female, *n* (%)	41	11 (26.8)	0.96 (0.42–2.20)	0.929		
Age (for 10 years of increase)			1.25 (0.91–1.71)	0.168		
Comorbidities, no. (%)						
Arterial hypertension	64	12 (18.7)	2.43 (1.08–5.46)	**0.031**		
Cardiovascular disease	33	15 (45.5)	3.13 (1.34–7.27)	**0.008**	4.56 (1.73–12.03)	**0.002**
Diabetes	20	7 (35)	1.54 (0.56–4.24)	0.405		
Obesity **^a^**	45	15 (33.3)	1.58 (0.71–3.50)	0.265		
Chronic renal disease **^b^**	12	7 (5.8)	4.40 (1.29–14.96)	**0.011**		
COPD	11	4 (4.4)	1.56 (0.43–5.79)	0.486		
Neoplasm **^c^**	10	1 (10)	0.28 (0.03–2.25)	0.228		
Neurological disorders	10	3 (33.3)	1.45 (0.39–5.40)	0.580		
NIV/OTI						
No NIV, yes OTI	40	17 (42.5)	1	**0.014**		**0.010**
Yes NIV, yes OTI	61	17 (27.9)	0.52 (0.23–1.21)		0.60 (0.25–1.46)	
Yes NIV, no OTI	27	1 (3.7)	0.05 (0.04–0.42)		0.04 (2.08–12.1)	

Abbreviations: IQR, interquartile range; COPD, chronic obstructive pulmonary disease; **^a^** obesity is defined as BMI > 30 kg/m^2^; **^b^** stage 3–5 of CKD, stages of chronic kidney disease; **^c^** solid neoplasia or hematological malignancy in the past five years; ICU, intensive care unit; OTI, orotracheal intubation; NIV, noninvasive ventilation; * At ICU 28-day follow up; OR, odds ratio; aOR, adjusted odds ratio. When *p*-value was <0.05 it was indicated in bold.

## Data Availability

The data presented in this study are available on request from the corresponding author. The data are not publicly available because of patient privacy and General Data Protection Regulation (GDPR).

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
