# Peer review of "Timing and Outcomes of Noninvasive Ventilation in 307 ARDS COVID-19 Patients: An Observational Study in an Italian Third Level COVID-19 Hospital"

_medicina, 2022, doi:10.3390/medicina58081104_

Round 1
Reviewer 1 Report
The manuscript presents an observational study of the outcome of non-invasive ventilation treatmnet on 307 ARDS COVID-19 patients. The study was conducted in an Italian third level COVID-19 hospital. The manuscript is well-written, the methodology is sound and the findings contribute to the scientific konwledge database. I therefore recommend acceptance.
Author Response
Dear Reviwer,
Thank you very much for appreciating our manuscript.
Kind regards,
All authors
Reviewer 2 Report
The purpose of the study was to examine the efficacy of NIV in COVID-19 patients and the factors associated with NIV failure. Although the aim of the study is interesting, there are major limitations which have to be mentioned.
Major comments:
1. Is the aim of the study clear?
Firstly, the retrospective design of the study does not permit firm associations with major outcomes, as mortality. In addition, there are some inconsistencies regarding the groups (eg inhomogeneous in NIV and non NIV group), the ARDS definition, and ‘’early stages’’ of COVID-19).
2. Is this a novel study?
There are several recent studies with a bigger sample size that have already examined the role of the NIV in COVID-19 patients. The sample size of this study, especially the severe ARDS group is quite small to have safe results.
Minor comments:
1. The settings of the NIV application are not mentioned, which play crucial role in the outcome of acute respiratory failure (eg CPAP, BiPAP, PEEP level).
2. How exactly did the authors measure PIP, plateau pressure, driving pressure during NIV?
3. In the inclusion criteria it is not defined the type of respiratory failure that the sample size experienced, an important parameter regarding the type of NIV that should be applied.
4. The group of patients who denied NIV or OTI should not have been included in the statistical analysis.
5. The 10 patients of the non NIV group with P/F ratio< 100 who did not underwent NIV or immediate intubation rises some concerns.
Author Response
Dear Reviewer,
we thank you very much for your valuable suggestions. We took your advice and made changes on the manuscript.
All revisions to the manuscript are flagged using the "Track Changes" function of MS Word, such that any changes can be easily viewed.
Below we will try to answer your questions as best we can:
The purpose of the study was to examine the efficacy of NIV in COVID-19 patients and the factors associated with NIV failure. Although the aim of the study is interesting, there are major limitations which have to be mentioned.
Major comments:
- Is the aim of the study clear?
Firstly, the retrospective design of the study does not permit firm associations with major outcomes, as mortality. In addition, there are some inconsistencies regarding the groups (eg inhomogeneous in NIV and non NIV group), the ARDS definition, and ‘’early stages’’ of COVID-19).
- Our study is not controlled nor prospective, but it is a retrospective observational study showing the full representativeness of real-life management of COVID-19 patients in our hospital during the pandemic. Since we considered all consecutive COVID-19 patients with pneumonia, the groups are not homogeneous as we can expect in controlled and prospective studies.
- Our retrospective observational study has to be considered a first step to formulating hypotheses that could incentivize the design of larger prospective studies.
- ARDS is defined fulfilling the Berlin criteria provided in Figure 1 of the Supplemental Material, and lines 113-114 of the revised manuscript.
- “early stages” of COVID-19 – with that we meant COVID-19 patients with P/F ratio > 200 mmHg – we replaced it with “P/F ratio > 200 mmHg“, provided in line 417 of the new manuscript.
- Is this a novel study?
There are several recent studies with a bigger sample size that have already examined the role of the NIV in COVID-19 patients. The sample size of this study, especially the severe ARDS group is quite small to have safe results.
"At the end of the paragraph of the Discussion, lines 439-461, we have highlighted in an extensive and clear way all the limitations of our study. Although this is not a novel study, there are still no clear recommendations on the field. It would be useful to be published more studies so that there could be enough evidence to develop reviews/good medical practice/recommendations by the scientific community. The novelty of our study, albeit with several limitations, is that it shows that starting NIV in COVID-19 patients with P / F> 200 mmHg appears to increase the efficacy of NIV and reduce the risk of orotracheal intubation or death."
Minor comments:
- The settings of the NIV application are not mentioned, which play crucial role in the outcome of acute respiratory failure (eg CPAP, BiPAP, PEEP level).
"Provided on lines 136-147 of the revised manuscript:
“The NIV included continuous positive airway pressure (CPAP) via helmet, Bous-signac mask, and bilevel positive airway pressure (BiPAP) via noninvasive ventilator machine (Philips Respironics Trilogy). We did not use high-flow nasal oxygen therapy (HFNOT) during the study period as the hospital was not provided. The first step of treating COVID-19 patients was 3-5 L/min oxygen administration through nasal googles or Venturi face mask with FiO2 up to 60% targeting a SpO2 level ≥ 94%, starting a monitoring of the vital signs. Patients with worsening respiratory status required an incremental respiratory support starting with CPAP with PEEP 7.5 cmH2O and FiO2 such that SpO2 ≥ 94%. Alternatively, in case of lack of CPAP devices, hypercapnic respiratory failure, risk of muscular exhaustion or further respiratory worsening, the support was switched at BiPAP starting with pressure support (PS) 5 cmH2O and PEEP between 5 and 10 cmH2O and gradually increasing according to the patient’s response.”
- How exactly did the authors measure PIP, plateau pressure, driving pressure during NIV?
"PIP was used as a parameter in patients in NIV and can be easily obtained by noninvasive devices. Plateau pressure and driving pressure were used as a respiratory parameter in patients in invasive mechanical ventilation and can be easily obtained by mechanical ventilator. – Provided on lines 159-162 of the revised manuscript."
- In the inclusion criteria it is not defined the type of respiratory failure that the sample size experienced, an important parameter regarding the type of NIV that should be applied.
"COVID-19 is typically associated with hypoxemic normo-hypocapnic respiratory failure. In the case of unresponsive hypercapnic respiratory failure, patients underwent orotracheal intubation and eventually in ECMO. Unfortunately, these data were not available for all subjects due to the difficulty of collecting this during the early phase of the pandemic."
- The group of patients who denied NIV or OTI should not have been included in the statistical analysis.
"One patient, who repeatedly refused orotracheal intubation, died while still in NIV. The 40 patients who underwent IOT without being treated with NIV were included in the statistical analysis to be able to compare with those who underwent NIV, Figure 3 and Table 3. – provided in lines 366-373 of the revised manuscript. “At hospitalization 307 (32.5%) COVID-19 patients fulfilled Berlin criteria for ARDS, and the latter had a statistically significant increased need for NIV treatment. In the non-NIV group 53 patients with moderate-severe ARDS (P/F ratio ≤ 200 mmHg) had contraindications to NIV (uncooperative patients, thoracic barotrauma, hemodynamic instability, refused NIV etc) or they were already severe ill when they were admitted to hospital. Indeed, 40 (5.6%) of the non-NIV group were admitted in ICU and subsequently intubated due to the worsening of their clinical conditions, of which 17 (42.5%) died within 28 days from ICU admission.”
- The 10 patients of the non NIV group with P/F ratio< 100 who did not underwent NIV or immediate intubation rises some concerns.
"The 10 patients with P/F ≤ 100 mmHg of the non-NIV group all underwent orotracheal intubation. These were patients who were admitted to our hospital, when they were already with severe ARDS at admission."
Please feel free to contact us if you have any further question.
Kind regards,
The authors

Reviewer 3 Report
Thank you for such interesting work and article; my comments are as follows;
1- it seems to me that 1st paragraph of introduction is not needed.
2-in the introduction part : positive end-expiratory pressure, PEEP to be written in such way: positive end-expiratory pressure (PEEP)
3-in the introduction part: pressure support, PS and fraction of inspired oxygen, FiO2 to be written in such way pressure support (PS) and fraction tion of inspired oxygen (FiO2).
4- In methods; page 3; I recommed to add for the exclusion criteria the other contraindications for NIV
5-In methods; page 3 line 122 (persistent low peripheral oxygen saturation, SpO2 < 92-94%.) please clarify the FiO2 given because higher than 60% over long period is not acceptable. In the same time, i would ask, If given FiO2 of 35% and the SPO2 was 92% , have you proceeded to NIV?
6- Results, first paragraph , i recommend not to duoblicate data presentation, i.e., whats is shown on the 1st figure in clear manner and % not to be rementioned on the text.
7- same comments; dublicated results on most of results text and related figures or tables.
Author Response
Dear Reviewer,
we thank you very much for your valuable suggestions. We took your advice and made changes on the manuscript.
All revisions to the manuscript are flagged using the "Track Changes" function of MS Word, such that any changes can be easily viewed.
Below we will try to answer your questions as best we can:
Thank you for such interesting work and article; my comments are as follows;
1- it seems to me that 1st paragraph of introduction is not needed.
We Removed it.
2-in the introduction part : positive end-expiratory pressure, PEEP to be written in such way: positive end-expiratory pressure (PEEP)
Done it, see line 78 of the revised manuscript
3-in the introduction part: pressure support, PS and fraction of inspired oxygen, FiO2 to be written in such way pressure support (PS) and fraction tion of inspired oxygen (FiO2).
Done it, see line 78 of the revised manuscript
4- In methods; page 3; I recommed to add for the exclusion criteria the other contraindications for NIV
Done it, see lines 149-152 of the revised manuscript
5-In methods; page 3 line 122 (persistent low peripheral oxygen saturation, SpO2 < 92-94%.) please clarify the FiO2 given because higher than 60% over long period is not acceptable. In the same time, i would ask, If given FiO2 of 35% and the SPO2 was 92% , have you proceeded to NIV?
Provided in line 133 and 140-147 of the revised manuscript: “The first step of treating COVID-19 patients was 3-5 L/min oxygen administration through nasal googles or Venturi face mask with FiO2 up to 60% targeting a SpO2 level ≥ 94%, starting a monitoring of the vital signs. Patients with worsening respiratory status required an incremental respiratory support starting with CPAP with PEEP 7.5 cmH2O and FiO2 such that SpO2 ≥ 94%. Alternatively, in case of lack of CPAP devices, hypercapnic respiratory failure, risk of muscular exhaustion or further respiratory worsening, the support was switched at BiPAP starting with pressure support (PS) 5 cmH2O and PEEP between 5 and 10 cmH2O and gradually increasing according to the patient’s response.”
In case of SpO2 of 92% with Venturi mask and FiO2 35%, the good medical practice was to gradually increase the FiO2 up to 60%, and in case of no response we proceeded to NIV.
6- Results, first paragraph , i recommend not to duplicate data presentation, i.e., whats is shown on the 1st figure in clear manner and % not to be rementioned on the text.
The percentages of the first paragraph has been removed, see lines 208-213 of the revised manuscript.
7- same comments; dublicated results on most of results text and related figures or tables.
Some duplicated data has been removed from the Results, see pages 6-7-8-9.
Please feel free to contact me if you have any further question.
Kind regards,
Authors

Round 2
Reviewer 2 Report
1. The stage of the COVID-19 disease is mainly characterized from the stage of the ARDS (early inflammatory, or late fibrotic) and an early stage is not necessary reflected from P/F ratio. Thus, it is of great importance characterization of the stage of the disease, as the NIV failure or success is strongly correlated to that, as we already know from previous published studies.
2. The positive effect of the NIV in mild ARDS (P/F ratio>200) of any origin is already well established from serial studies, and does not provide any novelty.
3. High Flow Nasal Cannula Therapy (HFNCT) in COVID-19 patients has been used with quite promising results in mild ARDS. Thus, as the study did not include this type of oxygenation, the comparison between NIV and conventional oxygen therapy provides limited information, and the results are not thoroughly estimated.
4. The authors have made changes in the sample size and statistical analysis, though the tables remain the same.